# Cellular and Molecular Mechanisms Underlying Synaptic Subcellular Specificity

**DOI:** 10.3390/brainsci14020155

**Published:** 2024-02-02

**Authors:** Mengqing Wang, Jiale Fan, Zhiyong Shao

**Affiliations:** State Key Laboratory of Medical Neurobiology and MOE Frontiers Center for Brain Science, Institutes of Brain Science, Department of Neurosurgery, Zhongshan Hospital, Fudan University, 131 Dong An Rd, Research Building B4017, Shanghai 200032, China

**Keywords:** synaptic specificity, subcellular compartment, secreted molecules, cell adhesion molecules, development

## Abstract

Chemical synapses are essential for neuronal information storage and relay. The synaptic signal received or sent from spatially distinct subcellular compartments often generates different outcomes due to the distance or physical property difference. Therefore, the final output of postsynaptic neurons is determined not only by the type and intensity of synaptic inputs but also by the synaptic subcellular location. How synaptic subcellular specificity is determined has long been the focus of study in the neurodevelopment field. Genetic studies from invertebrates such as *Caenorhabditis elegans* (*C. elegans*) have uncovered important molecular and cellular mechanisms required for subcellular specificity. Interestingly, similar molecular mechanisms were found in the mammalian cerebellum, hippocampus, and cerebral cortex. This review summarizes the comprehensive advances in the cellular and molecular mechanisms underlying synaptic subcellular specificity, focusing on studies from *C. elegans* and rodents.

## 1. Introduction

The human brain contains roughly 100 billion morphologically diverse neurons connected by over 100 trillion synapses. They are required for neural information storage and relays that are essential for locomotion, sensation, learning, memory, cognition, and other activities. Chemical synapses are one primary type of synaptic junction with asymmetric presynaptic and postsynaptic specialization [1]. At the presynaptic terminus, synaptic vesicles fuse with the presynaptic membrane and release neurotransmitters upon action potential, which is usually calcium dependent. The released neurotransmitters then bind to the corresponding receptors and activate or suppress the postsynaptic neurons [2,3,4,5]. As neurons have extremely complex morphologies, and each neuron forms thousands of synapses on average, the final output of any given neuron is determined by the sum of total synaptic inputs, which are affected not only by synaptic properties but also the physical location [6]. Therefore, to ensure proper neuronal information transmission and processing, specific presynaptic termini must precisely target specific postsynaptic subcellular domains of particular target cells [7,8,9,10,11]. Synaptic subcellular specificity is based on either presynaptic or postsynaptic locations depending on the model of study. For example, subcellular specificity was based on presynaptic localization in the *C. elegans* model and postsynaptic subcellular compartments in mammalian brain systems [9,11]. 

The synaptic specificity can be regulated at different steps during neurodevelopment, including neurogenesis, neuronal migration, axon guidance, synaptic targeting, synaptic formation, and synaptic pruning. How neurons select synaptic targets during development has been a fundamental question in neuroscience. Based on Peter’s rule, proposed by Peter and Feldman in 1976, synaptic connections are stochastically determined by the degree of anatomically overlapping of axons and dendrites [12]. Peter’s rule was supported by some studies [13,14]. However, as the spatial and temporal resolution increases, more and more evidence suggests that synapses are formed by nonrandom mechanisms [15]. In this review, we summarize comprehensive advances in regulating subcellular specificity in the synaptic targeting process, covering the roles of secreted/extracellular molecules, adhesion/transmembrane molecules, and cytoplasmic/intracellular molecules in the synaptic subcellular specificity. We also review the recent findings on the roles of neuronal activity and non-neuronal tissues in synaptic subcellular specificity.

## 2. Secreted Molecules

Many secreted signaling molecules, originally identified as guidance cues, are found to play multiple roles in neuronal development, including neurogenesis, cell migration, and synaptogenesis [16,17,18,19]. More recently, genetic studies in invertebrates demonstrate that many secreted molecules are required for synaptic subcellular specificity in both invertebrates [20,21,22,23] and mammals [24,25]. In this review, we focus on netrin/DCC, semaphorin/plexin, Wnt, and Slit/Robo, as they are involved in synaptic subcellular specificity.

### 2.1. Netrin

Netrin and netrin receptors were first identified in *C. elegans* (UNC-6/netrin and its receptors UNC-40/DCC and UNC-5) for their roles in behavioral coordination, cell migration, and axon guidance [26]. A few years later, their roles in *Drosophila* and vertebrate axon guidance were reported [27,28,29,30,31,32,33,34], suggesting that the functions of netrin and its receptors are highly conserved. Recent studies have demonstrated that netrin also regulates synaptic subcellular specificity [20,23,35,36,37].

In *C. elegans* heads, Amphid Interneuron Y (AIY) neurites anatomically can be divided into three zones. The ventral proximal Zone 1, the distal Zone 3 in the nerve ring, and the link Zone 2 are asynaptic, synaptic sparse, and synaptic enriched regions, respectively (Figure 1A,B) [20,38]. The synaptic specificity at the Zone 2 region is mainly determined by the UNC-6/netrin secreted from the ventral cephalic sheath glial cells (VCSCs), which contact AIY Zone 2 [20]. During embryogenesis, the VCSCs regulate the AIY synaptic subcellular specificity by expressing UNC-6/netrin [20]. Extending the VCSC glia to Zone 1 by genetic manipulation promotes ectopic synaptic formation in this region. This suggests that netrin acts as a short-range signaling molecule to regulate the synaptic assembly locally [20]. This specificity requires the AIY-expressed UNC-40/DCC receptor, which is specifically localized to the presynaptic sites [20]. In addition, UNC-6 and UNC-40 signaling also regulate the subcellular localization of serotonin neurosecretory–motor (NSM) neurons neurosecretory presynaptic terminals [36]. Furthermore, the dorsal axonic presynaptic localization of the ventral cord “dorsal A 9” (DA9) motor neuron in the tail is also regulated by the UNC-6/netrin signaling (Figure 1A) [37]. The DA9 dendrite and ventral axon accumulate more UNC-5 receptors than the dorsal axon, which receives the UNC-6 signals expressed in the ventral posterior side and locally inhibits the presynaptic assembly [37]. In *unc-6* or *unc-5* loss-of-function mutants, DA9 presynaptic proteins mislocalize to the ventral dendrite. Consistently, ectopic expressing UNC-6 in the presynaptic region is sufficient to exclude the endogenous synaptic localization. These data support that UNC-6 and UNC-5 localize the synaptic position by inhibiting the presynaptic assembly (Figure 1A) [37]. 

The role of UNC-6/netrins signaling in synaptic subcellular specificity is also reported in *Drosophila* [23]. The dendrites of A08a interneurons can be anatomically divided into medial and lateral regions, which receive inputs from dbd and A021 neurons, respectively. When overexpressing netrin receptor UNC-5 in the dbd neurons, it forms ectopic synapses on A08a lateral dendrites to avoid the netrins in the middle line region [23], suggesting the repulsive role of UNC-5 in synaptic subcellular targeting. Studies from both *C. elegans* and *Drosophila* suggest that netrin regulates synaptic subcellular specificity by promoting or inhibiting synaptogenesis depending on the receptors. The UNC-40 receptor is pro-synaptogenic, whereas the UNC-5 is anti-synaptogenic. Similarly, netrin-1 and DCC are involved in synaptic formation during development in vertebrate brains [35,39]. In frog embryos, injecting exogenous netrin-1 into the optical tectum promotes presynaptic formation rapidly in retinal ganglion cells [35]. In rodents, netrin-1 promotes the number and strength of excitatory synapses between cortical neurons in embryonic neuronal culture [39]. In the ventral tegmental area (VTA) of adult mice, netrin-1 loss-of-function in GABAergic or dopaminergic neurons leads to excitatory synapse reduction, suggesting the important roles of netrin-1 in synaptic maintenance [40]. Although their functions in synaptic subcellular specificity have not been reported, the conserved roles in pro-synaptogenesis in vertebrates suggest that netrin/DCC signaling may also regulate synaptic subcellular specificity in mammals. 

### 2.2. Semaphorins

Semaphorins are a class of proteins with a roughly 500 amino acid semaphorin domain, which are subdivided into eight subclasses based on their origins. The subclasses 1 and 2 are from invertebrates, 3–7 are from vertebrates, and the V is from virus [41]. The subclasses 2–3 and V are secreted, whereas the rest are membrane bound [41]. Semaphorins have a broad spectrum function in neurodevelopment, including neurogenesis, cell migration, and axon guidance [42]. Recent studies indicate that semaphorins are involved in synaptic subcellular specificity.

In *C. elegans*, axons of DA8 and DA9 motor neurons overlap in the dorsal nerve cord. However, each of them innervates a unique, non-overlapping muscle region [43]. This “synaptic tiling” phenotype is partially regulated by two transmembrane semaphorins SMP-1 and SMP-2 [22]. In SMP-1 mutants, presynaptic sites of DA8 and DA9 fail to tile, and they overlap. Interestingly, transmembrane SMP-1/SMP-2 and the receptor PLX-1 both act in the DA9 neuron to prevent the presynaptic sites overlapping between DA8 and DA9 through a cis interaction (Figure 1A). SMP-1/SMP-2 and PLX-1 are enriched at the asynaptic region tightly adjacent to the synaptic region to inhibit the synapse formation most likely through RAP-2/Rap2A GTPase, which regulates synaptic assembly through the effector MIG-15/TNIK (Figure 1A) [22,44]. 

Mammalian semaphorins also regulate synaptic subcellular specificity. Semaphorin3F (Sema3F), a secreted form of semaphorins, is required for the synaptic subcellular specificity in mouse dentate gyrus granule cells and neocortical pyramidal cells [25]. Specifically, mice without Sema3F display increased spine number and size in the approximal but not distal dendrites. Sema3F inhibits spine formation in the specific subcellular region by interacting with the holoreceptor Neuropilin2 and plexin A3 [25]. Transmembrane semaphorins also regulate synaptic formation and specificity. Sema5A, 5B, and 6A regulate the laminal-specific innervation in the inner plexiform layer [45]. Semaphorin4A and 4D promote, while semaphorin5A inhibits, synapse formation in hippocampal neurons [46,47,48,49]. These studies clearly demonstrate the conserved roles of semaphorins in regulating synaptic subcellular specificity.

### 2.3. Wnts

Wnts are highly conserved glycoproteins involved in many aspects of neurodevelopment and plasticity, including neurogenesis, axon guidance, and synaptogenesis [50,51]. The defects of the Wnt signaling are associated with many neurological disorders, including autism spectrum disorder (ASD), Alzheimer’s disease (AD), and Parkinson’s disease (PD) [52,53,54]. Wnts are also involved in synaptic subcellular specificity.

Studies in *C. elegans* demonstrate that Wnt signaling plays a complex role in presynaptic formation. Klassen and his colleagues found that the Wnt ligand LIN-44 secreted from the tail prevents synaptic formation at the DA9 posterior asynaptic region, which is mediated by localizing the frizzled receptor LIN-17 in that region. The frizzled receptor prevents local synaptic assembly by regulating the intracellular component disheveled DSH-1 (Figure 1A) [21]. The presynaptic subcellular specificity of DA8 motor neurons, whose synaptic sites are localized anteriorly next to DA9′s, is regulated by another Wnt, EGL-20 [55]. EGL-20, secreted from more anterior sites, locates a different frizzled receptor MIG-1 to the DA8 posterior asynaptic region, by which it inhibits the presynaptic assembly in the DA8 posterior region (Figure 1A) [55]. Therefore, two Wnt signaling pathways collaborate to regulate DA8/DA9 synaptic subcellular specificity by inhibiting synaptic formation at their posterior regions [21,55]. These data collectively demonstrate that Wnt signaling regulates synaptic subcellular specificity by inhibiting synaptic assembly. Interestingly, Wnts also play positive roles in synaptogenesis [56,57,58]. The Wnt ligand CWN-2 promotes synaptogenesis in AIY through a canonical Wnt signaling pathway [58].

In mammals, Wnt-7a expressed in granule cells promotes presynaptic assembly in cerebellum mossy fibers [59,60]. In the hippocampus, Wnt7a positively regulates dendritic spine development [61]. In contrast, Wnt5a has been showed to have both pro- and anti-synaptogenic functions. In mouse hippocampal tissue culture, Wnt5a activates a noncanonical signaling pathway and decreases active β-catenin levels, resulting in a reduction in the number of presynaptic puncta in hippocampal cultures [56]. In rat primary culture systems, Wnt5a promotes the recruitment of PSD-95 to the dendritic spine and regulates the dendritic spine morphogenesis [62,63]. The pro- and anti-synaptogenic roles of Wnts are probably due to the diversity and complexity of Wnt signaling and neurons.

Wnt signaling can also regulate synaptic subcellular specificity in combination with other factors. In *C. elegans*, loss of EGL-20/Wnt leads to axonal tiling defects in the DD5 and DD6 neurons, whereas presynaptic tiling is unaffected. However, when both EGL-20/Wnt and the gap junction protein UNC-9 are ablated, the synaptic tiling between these two neurons is lost [64].

### 2.4. Slit/Robo

Robo and the ligand Slits, first identified as a cue for preventing axons from recrossing midline in both *Drosophila* and mammals [65,66,67], are now found involved in developmental and pathogenic processes, including neurogenesis, angiogenesis, and cancer progression [68,69,70]. Recent studies find that Slits/Robo signaling is also involved in the synaptic subcellular specificity [23]. In *Drosophila*, Slit is highly expressed in the A08a midline domain, where low-Robo dbd neurons form synapses. When overexpressing robo-2 in dbd neurons, they will avoid the normal medial region and form synapses ectopically in the lateral region [23]. In mice, Robo2 is involved in establishing synaptic specificity in hippocampal CA1. Postsynaptically, Robo2 is present and necessary for the formation of excitatory (E) synapses, particularly in proximal dendritic compartments, with no impact on inhibitory (I) synapses and limited relevance to distal dendritic compartments [24].

Interestingly, these secreted synaptic subcellular specificity regulators also regulate axon guidance. Axon or dendrite guidance and synaptogenesis are sequential steps in neurodevelopment. Factors perturbating guidance will also affect synaptogenesis and specificity later. Therefore, it is challenging to determine whether the synaptic subcellular defects in the above-mentioned mutants happen during or after the axon guidance process. Future work should address this question.

## 3. Cell Adhesion/Transmembrane Molecules

Cell adhesion molecules (CAMs) are a group of cell membrane proteins regulating cell–cell and cell–extracellular matrix (ECM) interactions [71]. CAMs are recognized as key regulators in synaptic formation and organization [72,73,74]. Many of them are also involved in synaptic target recognition and subcellular specificity [75,76,77,78,79,80,81]. Therefore, they are often closely associated with various neurodevelopment and psychiatric disorders [82,83]. Here, we will focus on the role of immunoglobulin superfamily cell adhesion molecules (IgSF CAMs). Some other CAMs are discussed either with secreted (transmembrane semaphorins) or intracellular (latrophilin, teneurin, FLRT, and Nrg) regulators.

IgSF CAMs are a superfamily of adhesion molecules with various numbers of immunoglobulin domains. This family of CAMs plays an important role in synaptic development. For example, SynCAMs promote synaptogenesis in vitro through homophilic Ig domain interactions [84]; Sidekicks regulate laminar targeting in the inner plexiform layer during vertebrate retina development [85]. Recent studies have indicated that IgSF CAMs are involved in synaptic subcellular specificity in both invertebrates and vertebrates.

*C. elegans* hermaphrodite-specific neurons (HSNs) are located in the middle of the body close to the vulva, with a long process extending to the nerve ring in the head. The left HSN (HSNL) forms synapses specifically onto the vulval muscle interfaces. The presynaptic location specificity of HSNL is mediated by the heterophilic interactions between two IgSF CAMs SYG-1 and SYG-2 [86,87]. The ligand SYG-2 is expressed in vulval epithelial cells. It binds to and recruits SYG-1 in the HSNL axon at the specific subcellular sites and promotes local F-actin assembly. Then, F-actin promotes presynaptic assembly by interacting with its binding partner NAB-1 (Figure 1A) [88].

Several IgSF family L1CAMs have been found to regulate synaptic subcellular specificity in mice. In the cerebellum, basket cells form synaptic connections specifically onto the axon initial segment (AIS) of Purkinje cells [89]). This specificity is mediated by L1CAM neurofascin186 (NF186). NF186 forms an ankyrinG-dependent gradient along the AIS-soma axis of the Purkinje cell [89]. The NF186 expressed in AIS trans-synaptic binds to the Neuropilin-1 (NRP1) in basket cells to promote the synaptic assembly at the specific subcellular sites (Figure 2A) [90]. In the neocortex, L1CAM regulates the chandelier cells to form synaptic structure specifically onto the pyramidal AIS, which also requires ankyrinG in the pyramidal neurons [91]. In the cingulate cortex, L1CAM and ankyrin regulate basket cells to target onto the soma of pyramidal cells [92]. Disruption of L1CAM-ankyrin binding causes a significant decrease in the synaptic number between basket cells and the soma of pyramidal cells [92]. In addition, heterophilic interactions of different Ig family members between presynaptic and postsynaptic membranes are required for the synaptic assembly at specific subcellular locations [93]. For example, in the spinal cord, the presynaptic inhibitory axon–axon synapses are specified by two L1 family immunoglobulin NrCAM/CHL1 expressed in GABAergic interneurons and another IgSF CAM NB2 in the sensory neurons [93,94]. Therefore, the role of IgSF family CAMs in synaptic subcellular specificity seems very common in different species and cross CNS regions.

## 4. Intracellular Molecules 

The extracellular signaling functions through transmembrane proteins and intracellular factors to regulate synaptic assembly. Therefore, the final executors are intracellular factors. 

In *C. elegans*, CED-10/Rac1, MIG-10/Lamellipodin, CDC-42/CDC42, and PES-7/IQGAP are specifically localized to the AIY Zone 2 regions in response to extracellular signals mainly from VCSC glia, where they promote local presynaptic assembly [95,96]. Ectopic CDC-42 or PES-7 localization in the zone 1 region results in the ectopic presynaptic assembly (Figure 1B’) [95].

In the mouse cerebral cortex, presynaptic somatostatin-expressing (SST+) interneurons, parvalbumin-expressing (PV+) basket cells, and chandelier cells form synapses specifically on the distal dendrites, somata/proximal dendrites, and axon initial segments of pyramidal cells through specifically expressing *Cbln4*, *Lgi2*, and *Fgf13*, respectively [97]. Interestingly, *Cbln4* only promotes the synaptic formation when expressed in SST+, but not in basket cells or chandelier cells ([97], suggesting that the synaptic subcellular specificity regulated by *Cbln4* is cell-type specific.

Synaptic factor sorting and trafficking are required for the synaptic subcellular specificity. Local synaptic regulators have to be sorted and transported to the synaptic sites. Therefore, the sorting and trafficking of those synaptic regulators determine the synaptic subcellular specificity. In the mouse hippocampal CA1 region, the pyramidal neurons form regional-specific synaptic connections regulated by subcellular sorting synaptic adhesion molecules [98,99,100]. For example, the stratum oriens (SO) and stratum radiatum (SR) layers receive synaptic inputs from CA3 pyramidal neurons and the stratum lacunosum-moleculare (SLM) from the entorhinal cortex. This subcellular specificity is determined by subcellular specific sorting of the latrophilin-2/latrophilin-3 GPCR in the postsynaptic CA1 pyramidal neurons. The latrophilin-2 is localized specifically to SLM, whereas the latrophilin-3 is localized to SO and SR, where they promote synaptic assembly through binding to two presynaptic ligands teneurin and fibronectin leucine-rich repeat transmembrane proteins (FLRTs) (Figure 2B) [99]. Similarly, in the cerebral cortex, the pyramidal neurons sort neuregulin (Nrg)1 and Nrg3 in distinct subcellular domains, which mediates the inhibitory input and excitatory output synapses in the perisomatic and axonic regions (Figure 2C) [101].

While intracellular sorting and trafficking of many proteins is regulated by extracellular signaling, some are sorted intrinsically. In neuromuscular junctions, the synaptic subcellular specificity of the muscle cells is determined by the muscle-expressed MuSK, which is independent of the presynaptic motor neurons [102,103,104]. The MuSK is specifically localized to the central region of the muscle, which dictates the location of motor axons and synapses [105]. 

In addition to proteins, RNA sorting may also be involved in synaptic specificity. Although mRNA sorting has not been found to regulate synaptic subcellular specificity, its synaptic localization and the local translation are well-recognized [106,107,108]. Future work should focus on understanding the molecular mechanisms underlying the sorting and trafficking of local synaptic regulators.

## 5. Neuronal Activity-Dependent Mechanism

In addition to the intrinsic factors, environment-dependent neuronal activity can also regulate neurodevelopment, including proliferation, differentiation, migration, axon guidance [109,110,111,112], and synaptic plasticity [113,114,115,116,117,118,119,120,121]. Recently, neuronal activity has also been shown to regulate synaptic subcellular specificity [122,123].

Environmental temperature can affect neuronal activity and synaptic subcellular specificity. A recent study found that *C. elegans* AIY presynaptic specificity is regulated by the temperature-sensitive glutamatergic ASH neurons (Figure 1A) [123]. High cultivated temperature or overexpressing vesicle glutamate transporter EAT-4 induces ectopic synaptic formation in the AIY asynaptic Zone 1, which is mediated by a pair of presynaptic glutamatergic ASH neurons. Interestingly, these glutamate signals are sensed by a pair of inhibitory glutamate-gated chloride channels, GLC-3 and GLC-4, in the AIY interneurons [123]. The AIY synaptic defect induced by *eat-4(OE)* appears at the newly hatched larval L1 stage, suggesting that the glutamatergic neuronal activity regulates AIY synaptic subcellular specificity during embryogenesis, and it most likely regulates the synaptic formation, not the plasticity. Consistently, high-temperature treatment, specifically during the embryonic development stage, is sufficient to induce the synaptic subcellular specificity defects [123]. In mice, enriched environments have been shown to promote the expression of transcription factor NPAS4 in hippocampal pyramidal neurons, which specifically increases somatic inhibitory synapses through upregulating BDNF [122]. The above studies demonstrate that environmental conditions play an important role in synaptic subcellular specificity. 

## 6. Glia

Glia are closely associated with neurons and synapses, with profound effects on synaptogenesis, synaptic pruning and maturation, and synaptic subcellular specificity [89,124,125]. In *C. elegans*, as discussed in previous sections, the VCSC glia promote or maintain presynaptic structure at specific subcellular regions [20,126].

Glial cells also regulate synaptic subcellular specificity in vertebrates. In the mouse cerebral cortex, Bergmann glial (BG) fibers guide the GABAergic stellate axons towards Purkinje cell dendrites, which are mediated by the Close Homologue of L1(CHL1). CHL1 localized along BG fibers promotes stellate innervation of Purkinje dendrites [127]. Microglia also play important roles in synaptogenesis during adulthood. Reshef and his colleagues examined the effects of microglial depletion on the density and size of spines on the distal apical dendrites of adult-born granule cells (abGCs) within the external plexiform layer (EPL), where GCs form reciprocal synapses with mitral cells (MCs). They found that spine density was 25% lower in microglial depletion mice when compared with control mice [128]. The critical role of glia in specific synaptic pruning or specific neuronal compartments suggests their contribution to synaptic subcellular specificity [129,130]. Together, these findings indicate that glia have a general role in the modulation of synaptogenesis throughout the entire life span of animals.

## 7. Non-Neural Cells

During development, neural and non-neural cells are precisely coordinated. Therefore, communication between neural and non-neural tissues is critical for cell–cell and synaptic connection. Studies have demonstrated that non-neural tissues, including the epidermis, muscle, and intestine, are involved in synaptic development and specificity [37,55,58,87,126,131].

In the *C. elegans* nerve system, the presynaptic subcellular distribution in AIY interneurons is determined before hatching. Although animals increase up to 100-fold in volume during postembryonic development, the presynaptic pattern is maintained. The maintenance of AIY presynaptic pattern requires epidermal expressed the sialin homolog CIMA-1 specifically during the postembryonic developmental stage [126]. In *cima-1* loss-of-function mutants, presynaptic sites are normal in newly hatched animals. However, synaptic subcellular defects appear since the adult stage, which requires normal growth. The size of *C. elegans* is determined mainly by the epidermal- or muscle-expressed genes. Animals with dumpy morphologies, however, do not form the ectopic synapses in *cima-1* mutants [126]. Therefore, synaptic subcellular specificity is also affected by genes or molecules expressed in non-neuronal tissues, particularly those involved in body size growth. Muscle activity contributes to body elongation [132,133,134]. Interestingly, MIG-17, a secreted ADAMTS family metalloprotease, which is mainly expressed in the muscle tissue, regulates the AIY synaptic subcellular specificity [131]. Intestinal Wnt and neuropeptide signaling can also regulate synaptogenesis in the nervous system [58,135]. The above studies indicate that non-neuronal tissues also play a critical role in synaptic development and subcellular specificity. 

## 8. Conclusions and the Remaining Questions

Synaptic specificity is a prerequisite for normal neuronal functions and animal behaviors. With the increasing levels of resolution, the concept of synaptic subcellular specificity begins to be appreciated. Studies have shown that synaptic subcellular specificity is a conserved phenomenon present in both invertebrates and vertebrates. Through advanced imaging technology, we have gained some cellular and molecular insights. However, this is only the tip of the iceberg. Many questions remain to be addressed. First, the human brain has a myriad of types of neurons, and different types of neurons most likely use different sets of molecules to determine the synaptic subcellular specificity [8,11,136]. We only know very few of them so far. Secondly, although we have identified some molecules required for synaptic subcellular specificity, it is largely unknown how the synaptic subcellular border is determined. Thirdly, intracellular sorting of synaptic regulators is fundamental for synaptic subcellular specificity. However, it is largely unknown how subcellular sorting is regulated. Fourthly, we know very little about the role of synaptic remodeling at the subcellular level [137,138]. Finally, we have a lot to learn about the roles of non-neural tissues and neuronal activity in regulating synaptic subcellular specificity. Future studies should address those questions.

## Figures and Tables

**Figure 1 brainsci-14-00155-f001:**
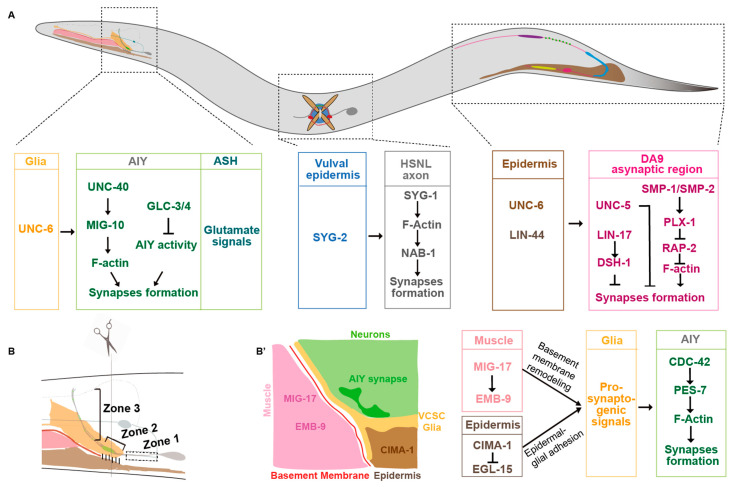
Presynaptic subcellular specificity in *C. elegans.* (**A**) A diagram of *C. elegans*. Some molecular mechanisms underlying the synaptic subcellular specificity in the head, middle body, and tail neurons are described. (**B**) A cartoon describes the anatomy of different tissues in the head (brown, epidermis; pink, muscle; red, basement membrane; yellow, VCSC glia; gray, AIY and ASH neurons; light green, nerve ring; green, AIY presynaptic sites. The AIY presynaptic pattern is stereotypic: the ventral asynaptic Zone 1 region (dashed box), the synaptic enriched Zone 2 region (skewed bracket), and the distal synaptic sparse Zone 3 region (vertical bracket). (**B’**) A cartoon describes the cross section at the AIY Zone 2 region. The molecular mechanisms underlying AIY synaptic subcellular specificity during embryogenesis are described in (**A**), whereas those during postembryonic development are described in (**B’**).

**Figure 2 brainsci-14-00155-f002:**
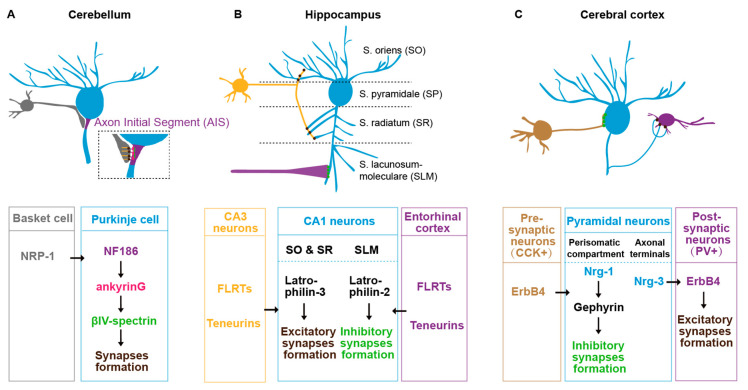
Postsynaptic subcellular specificity in mouse brain. (**A**) The synaptic subcellular specificity between basket cells (grey) and Purkinje cells (blue) axon initial segment (AIS) in mouse cerebellum. (**B**) The synaptic subcellular specificity of the CA1 pyramidal neurons (blue) in mouse hippocampus. The dendrites in the S. oriens (SO) and S. radiatum (SR) regions receive synaptic inputs from CA3 pyramidal neurons (yellow), whereas the dendrites in S. lacunosum-moleculare (SLM) receive synaptic inputs from entorhinal cortex (purple). (**C**) The synaptic subcellular specificity of pyramidal neurons (blue) in the cerebral cortex. Neuregulin (Nrg)1 and Nrg3 in pyramidal neurons are sorted into perisomatic and axonic domains, which are required for synaptic targeting from CCK neurons (brown) and to PV neurons (purple).

## Data Availability

No new data were created or analyzed in this study. Data sharing is not applicable to this article.

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
