# Peer review of "Cellular and Molecular Mechanisms Underlying Synaptic Subcellular Specificity"

_brainsci, 2024, doi:10.3390/brainsci14020155_

Round 1

Reviewer 1 Report

Comments and Suggestions for Authors

The proposed review deals with a subject that has not been published recently. It provides new examples from invertebrate models. However, the abstract should state that examples will be (mainly) taken from invertebrate (C. elegans and drosophila), otherwise it is somehow misleading. In addition the abstract mentions “spatially distinct subcellular compartment”, however, does it refer to pre- and postsynaptic compartments or to different compartments of the presynaptic neuron? Or both possibilities ? The same vagueness is present in the text.

The proposed review is very extensive, but limited to embryogenesis or mammalian prenatal development. While synaptogenesis still occurs during the postnatal period. Are the mechanisms describe in this MS still valid in adulthood?

Furthermore, readership from outside the C. elegans or Drosophila fields may not easily understand due to the large number of unexplained abbreviations used (AIY, NSM, zone 2, DA9 ...). Some of these are explained in the figure legend, but should also be defined in the text. What's more, relying solely on the text, without looking at the figure, makes it impossible to understand the organization of the worm's nervous system. This is essential, as the abstract does not state that the review deals primarily with invertebrates.

 Introduction, first paragraph:

-          Introduction should states that neurotransmitter release is calcium dependent, because it’s part of the definition of a neurotransmitter.

-          The specific receptor are not limited to the postsynaptic neurons, autoreceptors on the presynaptic neurons exist.

-          Synaptic size as a determinant of “the sum of the total synaptic inputs” is insufficient. Synaptic size and probability release and number of release site (active zone) per synapse (altogether building the synaptic weight) are important in addition to morphology and localization.

-          Do the “postsynaptic subcellular domains” refer to the synaptic and extrasynaptic localization of the receptor?

-          The sentence “synaptic subcellular specificity is based on either presynaptic or postsynaptic locations depending on the model of study” is used has a conclusion of the paragraph, whereas it should be an introduction and be explained and detailed later on.

Section “secreted molecules in synaptic subcellular specificity” should include subsection such as netrin, semaphoring, wnt, slits/robo … Otherwise transitions between the informal subsection have to be inserted.

Why UNC-6 epithelial expression is important ?

Explain the cis and trans conformation of plexins receptors.

Does asynaptic a specific neuronal zone in C. elegans or is it synonymous of extrasynaptic?

The MS states that wnt are highly conserved glycoprotein, however only the last sentence over 20 speaks of mammals, and in a very specific way? Moreover, the very last sentence of this paragraph should be more detailed.

The paragraph “secreted molecules can also regulate synaptic subcellular specificity in combination with …” is misplaced. It should be placed in the wnt subsection.

 Section “cell adhesion molecules in synaptic subcellular specificity” is interesting but instead of a list of molecule involved, it could have be organized in a more functional manner; excitatory vs. inhibitory synapse and somatodendritic vs. axoaxonic (AIS, synaptic bouton) synapse.

The last sentence “Heterophilic interactions of different Ig family …” is not a conclusion but calls for further details.

Section “neuronal activity-dependent mechanism”: the link between temperature and activity need to be explicated.

Section “non-neuronal roles in synaptic subcellular specificity”: the role of microglia in (adult) synaptogenesis is not described.

Author Response

Thank you for your suggestions, please see the attached point-to-point responses.

Reviewer 2 Report

Comments and Suggestions for Authors

I have reviewed the paper entitled “Cellular and molecular mechanisms underlying synaptic sub-cellular specificity” by Mengqing Wang, Jiale Fan and Zhiyong Shao.

In the abstract the authors state that the focus of this work is to study how synaptic subcellular specificity is determined in the neurodevelopment field and promise to summarize the comprehensive advances in the cellular and molecular mechanisms underlying synaptic subcellular specificity.

Indeed, a fine-tuned connectivity in neuronal circuits is a prerequisite for proper brain function and an interesting topic to perform future research.

Here are my comments:

Section 1- “Secreted molecules in synaptic subcellular specificity”

·        In the first paragraph the authors begin by saying: “Secreted signaling molecules play crucial roles in neuronal development (Dickson, 2002; Kolodkin and Tessier-Lavigne, 2011; Silhankova and Korswagen, 2007; Stoeckli, 2018). Many of them, including Netrin/DCC, Wnt, Slit/Robo-2, and Semaphorins, originally identified as guidance cues, are involved in synaptic specificity both in invertebrates and vertebrates.

Is there a reason why the authors chose to describe these 3 secreted molecules? The authors should add a sentence to introduce the reader that this section will provide information by focusing only in these molecules. Furthermore, the introduction in this section needs to be addressed more in depth. Please provide a broader introduction in this section. There is no reference cited for this last sentence. Please provide.

 ·        Second paragraph, “Netrin and the receptors were first identified in C. elegans (as UNC-6/Netrin and its receptors UNC-5 and UNC-40/DCC) for their roles in behavioral coordination, cell migration, and axon guidance (Hedgecock et al., 1990). “

This should read: Netrin and Netrin receptors….

·        Third paragraph. Before describing zone 2 the authors should provide information that the head has 3 zones to help the reader follow the paper. Not everyone is familiar with the anatomy of different tissues in the head or with the ventral asynaptic zone 1 region, the synaptic enriched zone 2 region, and with the distal synaptic sparse zone 3 region.

·        Please, provide a larger image of figure 1B. It is very difficult to distinguish the different zones (1-3) in the cartoon.

·        I suggest the authors to maintain the same order in the description (in this section) of the molecules (Netrin/DCC, Wnt, Slit/Robo-2, and Semaphorins ) that were first pointed out as involved in synaptic specificity. This order also helps the reader to follow the paper more easily.

·        Page 4: For Wnt's inhibiting synaptic formation many lines were provided. Only mentioning that Wnt also has pro-synaptogenic roles makes the paragraph seem unbalanced. Please expand this point.

 ·        Typo mistake “the pro- and anti-synaptogenic roles of Wnts are probably due to the diversity and complexity of Wnt signaling and neurons”.

-Start the sentence with capital letter.

·        I suggest moving this paragraph:

“Secreted molecules can also regulate synaptic subcellular specificity in combination with other factors such as gap junction proteins. In C. elegans, loss of EGL-20/Wnt leads to DD5 and DD6 neurons axonal tiling defect, while presynaptic tiling is unaffected. However, when both EGL-20/Wnt and the gap junction protein UNC-9 are both ablated, the synaptic tiling between these two neurons is lost (Hendi et al., 2022)”.

before the previous paragraph (Slits/Robo signaling) as a continuation of Wnt signaling.

·        Please, enlarge figure 2 fonts. They are difficult to read.

Author Response

(The authors gave the same response as above.)

Reviewer 3 Report

Comments and Suggestions for Authors

Here I present my comments on the Review entitled “Cellular and molecular mechanisms underlying synaptic subcellular specificity“ by Wang Mengqing, Fan Jiale, and Zhiyong Shao  for publication in Brain Sciences (Manuscript ID: brainsci-2765644)

This Review is well written and presented in a comprehensive manner. Also the figures are displayed clearly and in a comprehensive manner. The topic and propositions given by the authors are sound and respond to advances in the cellular and molecular mechanisms underlying synaptic subcellular specificity.

Although the work is in a good shape, still some addressed areas can be improved much more. Furthermore, other sections and concepts are poorly developed and they need to be improved to much the internal overall quality this work.

Initially, I felt enthusiastic when I saw interconnections of studies performed in C. elegans, Drosophila and mammal neurons, but later, somehow disappointed.  Most examples and arguments are build based on the invertebrate models while the argumentation based on mammalian neurons is much more restrictive. Evidence of this are, but not limited to, text such us:

Similarly, Netrin-1 and DCC are involved in synaptic formation and maintenance in mammals (Cline et al., 2023; Cramer et al., 2023).

 Similar to the secreted form, transmembrane semaphorins also regulate syn

aptic formation and subcellular specificity. Semaphorin4A and 4D promote synapse formation in mammalian hippocampus (Adel et al., 2023; Kuzirian et al., 2013; McDermott et al., 2018)

Wnts also play positive roles in synaptogenesis both in C. elegans and mammals (Davis et al., 2008; Park and Shen, 2012; Shi et al., 2018). the pro- and anti-synaptogenic roles of Wnts are probably due to the diversity and complexity of Wnt signaling and neurons.

These ideas, and other similarly stated, cry for a deeper analysis and test for supporting the concept of synaptic subcellular specificity.

 On the other hand, the contribution of glial cells on synaptic subcellular specificity, particularly in mammal models, is not dignified in the Review.I strongly encourage the authors to develop as a new independent section  the paragraph.Glial cells also regulate synaptic subcellular specificity in vertebrates. In mouse cerebral cortex, Bergmann glial (BG) fibers guide the GABAergic stellate axons towards purkinje cell dendrites, which is mediated by Close Homologue of L1(CHL1). CHL1 localized along BG fibers promotes stellate innervation of purkinje dendrites (Ango et al.,2008). The critical role of glia in specific synaptic pruning or specific neuronal compartments suggests their contribution to synaptic subcellular specificity (Baalman et al., 2015;

Favuzzi et al., 2021). Therefore, the role of non-neuronal cells in synaptic localization is most likely a general cellular mechanism underlying synaptic subcellular specificity.

 Comments on the Quality of English Language

Minor editing of English language required. I just recommend self-correcting typos and critically re-checking sentences/ideas that may be potentially misread.  

Author Response

(The authors gave the same response as above.)

Round 2

Reviewer 2 Report

Comments and Suggestions for Authors

I have now read the corrected version of the manuscript " Cellular and molecular mechanisms underlying synaptic subcellular specificity " and suggest that it should be:  Accepted in the present form.

Reviewer 3 Report

Comments and Suggestions for Authors

The authors have responded to my comments

Author Response

Thank you so much for your time and comments.